# Vaccine Acceptance and Hesitancy among Hospitalized COVID-19 Patients in Punjab, Pakistan

**DOI:** 10.3390/vaccines10101640

**Published:** 2022-09-30

**Authors:** Mohamed A. Baraka, Muhammad Nouman Manzoor, Umar Ayoub, Reem M. Aljowaie, Zia Ul Mustafa, Syed Tabish Razi Zaidi, Muhammad Salman, Chia Siang Kow, Mamoon A. Aldeyab, Syed Shahzad Hasan

**Affiliations:** 1Clinical Pharmacy Program, College of Pharmacy, Al Ain Campus, Al Ain University, P.O. Box 64141, Abu Dhabi, United Arab Emirates; 2Clinical Pharmacy Department, College of Pharmacy, Al-Azhar University, Cairo 11651, Egypt; 3Department of Medicine, Tehsil Headquarter (THQ) Hospital, Bhakkar 30000, Pakistan; 4Department of Medicine, Tehsil Headquarter (THQ) Hospital, Fortabbas 62020, Pakistan; 5Department of Botany and Microbiology, College of Science, King Saud University, P.O. Box 2455, Riyadh 11451, Saudi Arabia; 6Discipline of Clinical Pharmacy, School of Pharmaceutical Sciences, Universiti Sains Malaysia, Penang 11800, Malaysia; 7Department of Pharmacy Services, District Headquarter (DHQ) Hospital, Pakpattan 57400, Pakistan; 8HPS Pharmacies, Institutional Care, EBOS, Melbourne, VIC 3052, Australia; 9School of Healthcare, University of Leeds, Leeds LS2 9JT, UK; 10Faculty of Pharmacy, University of Lahore, Lahore 54660, Pakistan; 11School of Pharmacy, International Medical University, Kuala Lumpur 53200, Malaysia; 12Department of Pharmacy, University of Huddersfield, Huddersfield HD1 3DH, UK

**Keywords:** COVID-19, Pakistan, SARS-CoV-2, vaccine acceptance, vaccine hesitancy

## Abstract

Vaccine hesitancy is widespread in many parts of the globe, particularly in low–middle-income countries. Therefore, we surveyed a sample of hospitalized COVID-19 patients to assess COVID-19 vaccine acceptance and vaccine hesitancy in a low–middle-income country. A cross-sectional sample of 385 confirmed reverse transcriptase–polymerase chain reaction COVID-19 patients treated at secondary and tertiary care hospitals in Punjab, Pakistan, were analyzed to assess COVID-19 vaccine uptake and vaccine hesitancy. The construct validity and reliability of the 11-item vaccine hesitancy questionnaire were also examined. In addition, multivariate logistic regression was used. The majority of the COVID-19 patients admitted to hospitals were not vaccinated (84%). Of those who were willing to receive vaccination, the majority (55%) considered vaccines an effective way to protect people from COVID-19. However, those who were not willing to receive their COVID-19 vaccine had significantly higher hesitancy than those willing to receive their COVID-19 vaccine. In addition, older hospitalized COVID-19 patients aged 60 years or above (20–29 years: OR 0.10; 95% CI 0.01–0.72, *p* = 0.001) and patients from urban areas (OR 3.16 95% CI 1.27–7.87, *p* = 0.013) were more likely to receive the COVID-19 vaccine than younger patients and patients from rural areas. Patients with no formal education had significantly higher hesitancy (OR 5.26; 96% CI 1.85–14.97, *p* = 0.002) than participants with graduation and above education. More than half of the study’s participants did not trust information shared on social media about COVID-19 vaccines and cited newspapers/news channels as their main source of information. The study provides important insights into COVID-19 vaccine acceptance and the impact of vaccination campaigns. Many unvaccinated COVID-19 patients in hospitals highlight the need for an effective vaccination drive to protect people from acquiring infection and subsequent hospitalization.

## 1. Introduction

The discovery and development of various vaccines against coronavirus disease 2019 (COVID-19) have transformed the response of all governments toward the COVID-19 pandemic. Nationwide lockdowns are increasingly becoming a thing of the past, and many countries are contemplating unrestricted movements for vaccinated individuals. Despite this, the world is witnessing intelligent, calculated, and, most importantly, falsified misinformation campaigns against the available vaccines and mass vaccination programs against COVID-19. The movements against vaccines are notoriously famous for spreading misinformation about the rigorously investigated vaccines.

Pakistan is a culturally diverse country wherein health beliefs are often intermingled with cultural and political views. The continual existence of poliomyelitis and tuberculosis is a testament to significant vaccination hesitancy or disbelief against two otherwise widely eradicated infectious diseases. One-third of Pakistani children have not received routine childhood immunization, with the immunization rates as low as 37% in some of the rural areas of Pakistan [1].

Pakistan has been fortunate to have a wide range of COVID-19 vaccines available for public use. To date, the BBIBP-CorV vaccine (Sinopharm), CoronaVac vaccine (Sinovac), Ad5-nCoV vaccine (CanSino), mRNA-1273 vaccine (Moderna), ChAdOx1-S vaccine (AstraZeneca), BNT162b2 vaccine (Pfizer-BioNTech), Gam-COVID-Vac vaccine (Sputnik V), and Pakistani-made Pakvac vaccines have been approved for use in the COVID-19 national immunization campaign [2]. The government of Pakistan ran the first formal COVID-19 vaccination campaign on 2 February 2021. In the first phase of the vaccination drive, the front-line healthcare workers had been prioritized to receive COVID-19 vaccines free of cost [3]. With the establishment of multiple vaccine centers in every district, older individuals over 60 years received free-of-charge COVID-19 vaccines in the next phase [4].

In order to avoid high morbidity, mortality, and economic catastrophe and the unusual burden on the healthcare delivery system caused by the COVID-19 pandemic, the majority of the country’s population should acquire immunity through a comprehensive vaccination campaign [5]. A successful vaccination campaign requires addressing fundamental queries of the masses related to vaccine safety and efficacy to increase their willingness to receive COVID-19 vaccines. Currently, all individuals older than 18 years can receive their COVID-19 vaccines by visiting any nearby vaccine center related to the Government of Pakistan [6]. Unfortunately, recent studies from Pakistan have highlighted that misbeliefs and doubts of effectiveness were associated with vaccine hesitancy [7,8]. While the concentrated vaccination campaigns are backed up by print and electronic media, in addition to the ongoing mobile health messaging services via mobile phone operators, only about 43% of Pakistan’s total population received two doses of the COVID-19 vaccines, whereas 56% received at least one dose of the COVID-19 vaccines [9].

The low vaccination rate is not surprising considering the widespread confusion and misinformation campaigns. As most of the available COVID-19 vaccines are relatively new, campaigns against COVID-19 vaccines can be easily operated by campaigners who are against the vaccines. Countries with low literacy rates and poor socioeconomic status are particularly vulnerable to campaigns against vaccines due to poor health literacy and the lack of trust toward the established professional and governmental bodies’ among their citizens. There have been many public concerns about the safety of COVID-19 vaccines since the beginning of this pandemic, contributing to vaccine hesitancy [10]. Vaccine hesitancy has been defined as the delay in accepting or refusing a vaccine, although vaccination services are made available [11]. This has been a global problem wherein vaccine hesitancy is widespread in many parts of the globe, particularly in low–middle-income countries (LMICs), due to unchecked, misleading, and false information spreading through social media platforms [7,12,13]. Vaccine hesitancy and its lower acceptance have been reported among healthcare workers and the general population of Pakistan due to safety concerns, the lack of appropriate vaccine efficacy, misbeliefs, etc. [7,8,14,15]. However, if the majority of the general public are unvaccinated, there would be a higher possibility of a large pool of virus hosts, replication, transmissibility, and antigenicity [16]. This study aimed to assess vaccination status and vaccine hesitancy among hospitalized patients with COVID-19 in the largest province of Pakistan using a novel study instrument.

## 2. Materials and Methods

### 2.1. Study Design and Population

This was a cross-sectional study which utilized an online interviewer-administered tool among patients admitted to COVID-19 wards (between 1 August 2021 and 30 September 2021) with a positive real-time reverse transcriptase–polymerase chain reaction (RT-PCR) test in four district headquarters hospitals and three tertiary care hospitals in Punjab, Pakistan. Patients who were at least 18 years of age, of any gender, admitted to the COVID-19 wards with a positive RT-PCR test, and who provided consent were invited to participate in the present study. The survey was administered by healthcare professionals providing care to the hospitalized COVID-19 patients.

### 2.2. Development and Validity of the Survey Instrument

The questions included in the study instrument were developed based on the input received from healthcare providers involved in COVID-19 vaccination and the factors potentially associated with vaccine hesitancy.

The study instrument comprised three sections. The first section of the study instrument included questions about demographic information (e.g., gender, age, long-standing illness, and geographical location). The second section of the study instrument was designed to collect information about the vaccination status of the participants (e.g., who received the COVID-19 vaccine, any adverse effects, family members who received vaccines, etc.). Finally, the last part of our instrument had a new questionnaire measuring vaccine hesitancy among hospitalized COVID-19 patients. Vaccine hesitancy is defined as a delay in accepting or refusing safe vaccines despite the availability of vaccine services [12,17]. The initial draft of the 15-item study instrument measuring vaccine hesitancy was developed by the senior author (SSH), which was later subjected to content validity. The instrument was shared with four experts—two academicians (one with an epidemiology research background and the other with expertise in psychometric testing of questionnaires) and two healthcare professionals (medical practitioners with research experience). The experts were given one week to comment on the content of the questionnaire before sending their feedback. This step was carried out to ensure that the study instrument represents all facets of a given construct. A final version of the study instrument (11-item) was produced after incorporating changes based on the comments received from the reviewers. The instrument was piloted among a small group of patients to ensure the instrument was clear and feasible to use [18]. Construct validity (convergent and discriminant validity) and reliability analyses were also performed. Each of the eleven items was with 5-point response options (strongly agree, agree, not sure, disagree, or strongly disagree) and were summed (score range: 11 [agreement]-55 [disagreement]); higher score indicates higher vaccine hesitancy.

The English version of the study instrument was translated into Urdu with forward–backward translation [19]. Translating the study tool into a local language maintains consistency among the healthcare professionals involved in the data collection process. The healthcare professionals performed the forward translation of the instrument into the Urdu language. Urdu was their first language, and English was their working or official language. They also possessed knowledge of the health concepts used in the study tool. The backward translation followed the same approach. The conceptual and cultural equivalence was the core of this exercise instead of linguistic equivalence.

### 2.3. Sample and Sampling

The sample size was estimated using the total number of COVID-19-positive people when this survey was conducted in the Punjab province of Pakistan. The sample size of 383 was estimated, considering a 5% margin of error, a 95% confidence interval, and a 50% response distribution. We invited all consecutive patients with a positive RT-PCR result admitted to the COVID-19 wards in four district headquarters’ hospitals and three tertiary care hospitals. Due to the COVID-19-related restrictions in the participating hospitals, an online, web-based version of the study tool, with an Urdu translation, was developed and used by healthcare professionals involved in data collection. These healthcare professionals (involved in the research project) accessed the online form from mobile phones or other devices. They asked questions to the patients (interviewer-administered) after explaining the study objectives and receiving verbal consent from the patients.

### 2.4. Statistical Analysis

Data are presented as frequencies, percentages, mean/median, and standard deviation. Since a new vaccine hesitancy questionnaire has been developed, the construct validity was examined using the exploratory factor analysis (EFA) method. EFA method was used with Kaiser’s alpha factoring and rotated using varimax orthogonal rotation, which improves the solution compared to unrotated ones and allows factors to be independent of one another. The sample adequacy was measured using the Kaiser rule (Kaiser–Meyer–Olkin test (KMO)) and an eigenvalue of more than one [20,21]. In addition, Bartlett’s test of sphericity measured significant correlations between variables. Convergent, that is, when items loaded highly on their factors and discriminant validity, the presence of cross-loadings, and/or strong correlations between factors (factor loadings of more than 0.75) were also assessed. The internal consistency or reliability of the study questionnaire was determined using Cronbach’s alpha (α) [22], where the alpha coefficient determines the extent to which multiple indicators belong together for a latent variable [23]. A commonly accepted threshold for reliability is more than or equal to 0.70. However, values below 0.70 are also acceptable [24,25]. A chi-squared test was used to compare the vaccinated and non-vaccinated groups for their demographic characteristics and vaccination-related factors. Multivariate logistic regression with a backward stepwise model examined the association between vaccination status and hesitancy with participants and vaccine-related factors. The following variables were entered into the model: age, gender, education, occupation, marital status, geographical location, long-standing illness, received information about the vaccine, family members, and overall hesitancy. The model was selected based on the model summary with a Hosmer and Lemeshow test. The Statistical Package for Social Sciences (SPSS^®^ version 27) was used, with a *p* = 0.05 indicating a significance level.

## 3. Results

### 3.1. Demographic Characteristics of COVID-19 Patients by Vaccination Status

A total of 385 COVID-19 patients admitted to COVID-19 wards in secondary and tertiary care hospitals participated in this study. Only a small proportion of COVID-19 patients admitted to hospitals were vaccinated (63, 16.3%) and either received one or two doses of COVID-19 vaccines. On average, vaccinated patients were slightly older than unvaccinated patients (47.2 years versus 50.4 years) (Table 1). Among the vaccinated group, most COVID-19 patients were females (54%), attained secondary education (44%), aged 40 years or above (87.3%), and were from urban areas (68.3%). Except for the geographical location, vaccinated and non-vaccinated had similar characteristics (*p* > 0.05).

### 3.2. COVID-19 Vaccination-Related Factors by Vaccination Status

Interestingly, the majority of the family members or friends of the study participants had received their COVID-19 vaccines (307, 79.7%). More than three-fourths of the family members or friends of vaccinated and non-vaccinated COVID-19 patients had received their COVID-19 vaccine doses (Table 2). In addition, nine out of ten vaccinated or non-vaccinated COVID-19 patients had received information about COVID-19 vaccines (92%), whereas six out of ten used newspapers/news channels as their main source of information (61%). Higher number of non-vaccinated patients (89%) used social media than vaccinated patients (11%) to receive COVID-19 vaccine-related information. However, more than half of the study participants (81% unvaccinated patients and 19% vaccinated patients) had no trust in information shared on social media about the safety and efficacy of COVID-19 vaccines.

### 3.3. Construct Validity and Reliability of Vaccine Hesitancy Questionnaire

The KMO test of sampling adequacy revealed an overall index of 0.94, indicating that the sample was adequate for factor analysis. Bartlett’s test of sphericity revealed that the inter-correlation matrix was factorable (chi-square (55) = 6069.91, *p* < 0.001) (Appendix A). The study results showed acceptable convergent and discriminant validity. All 11 items had high factor loadings (convergent) with some cross-loading (discriminant) between factors (Appendix A). The EFA produced a two-factor solution (two constructs) for the items in the questionnaire; the total variance explained by two factors was 85.8%. Based on the results, 76.4% of the variance was explained by the first factor, labelled ‘hesitancy in receiving COVID-19 vaccine,’ while the second, labelled ‘trust in COVID-19 vaccine,’ explained 9.4% (Table 3). The overall reliability (α) value of the survey was 0.97, indicating excellent reliability. Both factors were excellent indicators of the dimensions they represent as the reliability of each construct was considered adequate, that is, more than 0.90.

### 3.4. Patients’ Willingness to Receive COVID-19 Vaccine and Hesitancy

Figure 1 presents the willingness to receive the COVID-19 vaccine by patient-related factors. Unvaccinated participants living in rural or sub-urban areas, people with more formal education, and people 60 years of age were more reluctant (not willing) to receive the COVID-19 vaccine. Table 4 presents the demographics of unvaccinated COVID-19 patients by their willingness to receive their vaccine doses. Except for gender, all patients-related factors were significantly different in terms of their willingness to receive vaccine. Among the unvaccinated patients, younger patients (<50 years), patients with secondary or above education, employed patients, patients living in urban areas, and patients with long-standing illnesses were more willing to receive their COVID-19 vaccine in future than their counterparts. Of 322 unvaccinated patients, 151 (46.9%) patients were categorized as vaccine hesitant.

### 3.5. Factors Associated with Vaccine Acceptance and Hesitancy

Figure 2 presents the mean hesitancy score with standard deviation by patient-related factors. Patients 60 years or above and patients with no formal education were found to have higher hesitancy scores than their counterparts. Table 5 presents the results of multivariate logistic regression. Older hospitalized COVID-19 patients aged 55 or above had significantly higher odds of receiving the COVID-19 vaccine than their younger counterparts (20–29 years: OR 0.10; 95% CI 0.01–0.72, *p* = 0.001). Males had significantly lower odds of receiving the COVID-19 vaccine (OR 0.54; 95% CI 0.30–0.96, *p* = 0.037) and had higher hesitancy (OR 3.56; 95% CI 1.67–7.59, *p* = 0.001) than females. Regarding geographical location, COVID-19 patients from urban areas (OR 3.16; 95% CI 1.27–7.87, *p* = 0.013) or sub-urban areas (OR 3.88; 95% CI 1.35–11.15, *p* = 0.012) had significantly higher odds of receiving the COVID-19 vaccine than patients from rural areas. Compared to patients with an education attainment of graduation and above, patients with no formal education had significantly higher hesitancy (OR 5.26; 96% CI 1.85–14.97, *p* = 0.002).

## 4. Discussion

The present study explored the vaccination status and vaccine hesitancy among hospitalized COVID-19 patients and examined the factors associated with vaccine uptake and hesitancy in Pakistan. The current study revealed a low COVID-19 vaccine uptake (16%) among COVID-19 patients admitted to COVID-19 wards in four districts and three tertiary care hospitals in Punjab, Pakistan. Similarly, a previous study from LMICs indicated that the acceptance of COVID-19 vaccines in Pakistan is reported to be the lowest among the LMICs [13,26]. Moreover, previous immunization records indicated that Pakistan could not achieve the vaccination targets of Bacillus Calmette Guerin (BCG), measles, and polio due to a lack of trained health professionals, logistics, and education with religious beliefs [15].

The low uptake level of COVID-19 vaccines in our cohort of patients could be due to multiple factors, and we believe that some of these factors are common in the global community. Firstly, uncertainties exist regarding the safety of the COVID-19 vaccines, especially vaccines that utilized mRNA technology, since there have been no prior experience or successes. Secondly, there could be doubts over the efficacy of the COVID-19 vaccines since there have been reports of infections in those who have been fully vaccinated. Thirdly, numerous campaigns launched by those against vaccines undermine people’s trust in COVID-19 vaccines. People against vaccination often utilize social media to disseminate fabricated, false, and sometimes misleading news to achieve their agenda. Nevertheless, the present study identified important factors associated with vaccine hesitancy, such as being younger than 55 years of age and originating from rural areas. Therefore, to achieve enough immunization coverage against COVID-19, populations from the above groups can be specifically targeted to improve their acceptance of the COVID-19 vaccine. The factor that could increase vaccine hesitancy, especially among the rural population, is a lack of education or access to quality education. In our study, patients with no formal education showed higher vaccine hesitancy compared with patients with secondary or above education. The low literacy rate in Pakistan coupled with easy access to unauthentic information is a major concern that should be addressed to improve future vaccine uptake in the region.

Interestingly, in our study, we noted that males were significantly less likely to accept the vaccine. Such a finding is in contrast with the findings of several other studies [27,28,29,30] performed in LMICs. Indeed, as reported in a systematic review and meta-analysis [31], a majority (58%) of the included studies reported men to have higher intentions of becoming vaccinated against COVID-19, and that significantly fewer women stated that they would receive vaccination than men (OR 1.41; 95% CI 1.28 to 1.55). Lower vaccination intentions among men could be problematic for various reasons. Pakistani society is considered to be a male-dominant society wherein men are the primary authority figures and women are subordinate. Therefore, at the household level, men may negatively influence the willingness of family members to receive COVID-19 vaccines. Moreover, these Pakistani men may expose themselves to the danger of COVID-19 since men have been reported to suffer from a more severe disease and higher mortality during the global COVID-19 pandemic. 

One encouraging finding was that more than 60% of the unvaccinated study population in our study are willing to receive their vaccines to protect themselves against COVID-19 infection. Another study published in Ethiopia demonstrated that a similar proportion of the non-vaccinated study population was willing to receive the vaccine [32]. Our findings are somewhat promising compared to a previous study conducted by Piltch et al. in the US, revealing that nearly 40% of the survey population were planning or willing to receive COVID-19 vaccines [33]. More than half (54.7%) of the non-vaccinated population were willing to take COVID-19 vaccines because they considered it a safe way to protect themselves from being infected with COVID-19. Our study results are similar to the previous study conducted among the general public in Pakistan [15], wherein the majority of the study’s participants showed a positive attitude towards the safety of COVID-19 vaccines to prevent the spread of the infection. Moreover, more than half of the non-vaccinated COVID-19 patients in our study were willing to receive the COVID-19 vaccine because of the health concerns of their families and friends. These findings are similar to the conclusions of a previous study in which the health matters of family and friends were positively associated with the acceptance of COVID-19 vaccines [15]. Around 40% of the study population reported higher hesitancy toward the COVID-19 vaccine, and 60% reported lower hesitancy over the vaccine type provided by the government. These findings corroborate the previous studies conducted in the United States, United Kingdom, and Australia, highlighting that most study participants trust their national government and have higher COVID-19 vaccine acceptance [34].

Due to the highly contagious nature of COVID-19 and poor health infrastructure in Pakistan, this study reinforces the importance of the promotion of COVID-19 vaccines at all levels as early as possible to reduce the risk of infection, which could consume already limited healthcare resources. This is especially urgent given the concerns that the new virus variants are emerging. In addition, numerous strategies can effectively accelerate vaccination administration among hesitant population groups [35]. For example, healthcare professionals can play a key role in encouraging unvaccinated COVID-19 patients admitted to the hospital and arrange a follow-up visit after discharge to vaccinate these people. In addition, the religious leaders of the country can communicate the importance of being vaccinated to the community, including sharing stories of their own vaccinations. Moreover, the Pakistani government can collaborate with and support community organizations to utilize existing structures to educate and vaccinate the Pakistani population.

There are a number of limitations that could affect the generalizability of this study’s findings. A few limitations are inherent in our study as we used a convenient sampling technique to recruit study participants that may be associated with some degree of selection bias. Moreover, the hospitalized COVID-19 patients from only seven hospitals in the province of Punjab were approached by the investigators in this study to extract necessary information from them and to record their responses; therefore, we should be careful when considering generalizing our results. Although we have adjusted for confounders, other factors could affect the findings (e.g., illiteracy, poor access to health-related information, etc.). Moreover, we did not attempt to compare trust levels among sample subgroups. In this study, the information about hesitancy and potential confounders was based on the self-report instrument. Therefore, there is possibility for information bias and the misclassification of the vaccination status. Despite the limitations, we are confident that the findings of our study provide baseline and robust information about vaccine hesitancy among hospitalized COVID-19 patients and highlight the importance of tackling this problem to prevent spreading the infection and exhausting the health-related resources.

In conclusion, the present study provides important insights into vaccination status and the impact of a vaccination campaign in Pakistan. Less than a quarter of hospitalized COVID-19 patients have received COVID-19 vaccine doses. Interestingly, vaccinated patients were more hesitant to receive COVID-19 vaccines than non-vaccinated patients. This hesitancy could be due to their negative experience of acquiring COVID-19 infection and subsequent hospitalization after receiving their COVID-19 vaccine doses. This could also discourage other non-vaccinated people from accepting the COVID-19 vaccine. Our findings may guide the efforts to strengthen the COVID-19 vaccination campaign or any other vaccination campaigns in the future.

## Figures and Tables

**Figure 1 vaccines-10-01640-f001:**
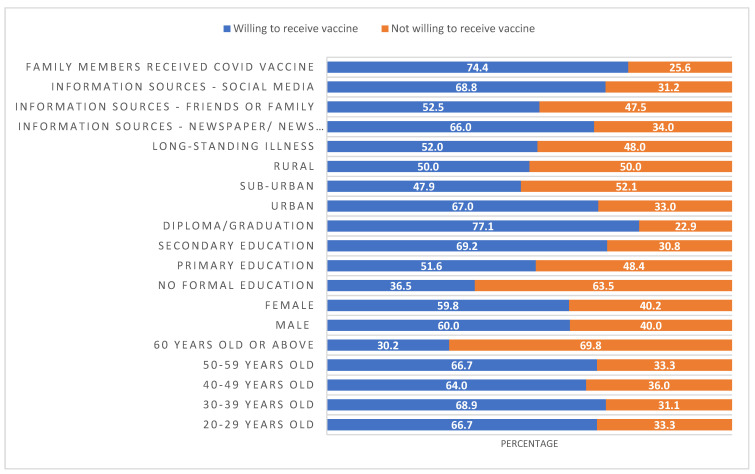
Percentage of unvaccinated participants willing or not willing to receive the vaccine in each of the patient-related categories.

**Figure 2 vaccines-10-01640-f002:**
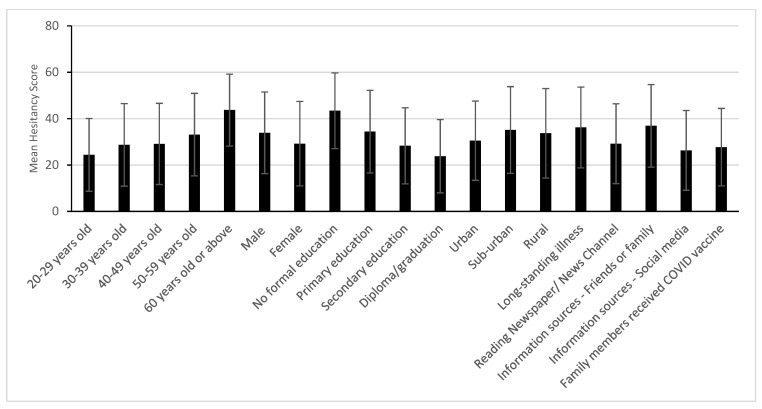
Comparison of hesitancy score (using mean and standard deviation) between patient-related factors among unvaccinated patients. Note: score range: 11–55; the higher the score, the more hesitant the person.

**Table 1 vaccines-10-01640-t001:** Demographic characteristics of COVID-19 patients by vaccination status.

Variables	Non-Vaccinated COVID-19 Patients *n* (%)	Vaccinated COVID-19 Patients *n* (%)	*p*-Value
**Age, mean (SD)**	47.2 (12.8)	50.4 (11.4)	0.063
**Age Groups**			
20–29	33 (91.7)	3 (8.3)	0.067
30–39	45 (91.8)	4 (8.2)	
40–49	111 (81.0)	26 (19.0)	
50–59	81 (82.7)	17 (17.3)	
60 or above	52 (80.0)	13 (20.0)	
**Gender (*n* = 384)**			
Male	189 (86.7)	29 (13.3)	0.060
Female	132 (79.5)	34 (20.5)	
**Education (*n* = 385)**			
No formal education	73 (86.9)	11 (13.1)	0.443
Primary	62 (81.6)	14 (18.4)	
Secondary	117 (80.7)	28 (19.3)	
Diploma/Graduation	70 (87.5)	10 (12.5)	
**Occupation (*n* = 384)**			
Unemployed	52 (16.2)	7 (11.1)	0.479
Self-employed	91 (28.3)	17 (27.0)	
Employed	103 (32.1)	19 (30.2)	
Retired	75 (23.4)	20 (31.7)	
**Marital status (*n* = 384)**			
Unmarried/never married	29 (90.6)	3 (9.4)	0.330
Married	227 (81.9)	50 (18.1)	
Separated/divorced/widowed	65 (86.7)	10 (13.3)	
**Geographical location (*n* = 384)**			
Urban	194 (81.9)	43 (18.1)	0.022
Sub-urban	48 (77.4)	14 (22.6)	
Rural	79 (92.9)	6 (7.1)	
**Long-standing illness (*n* = 384)**			
No	199 (84.7)	36 (15.3)	0.470
Yes	122 (81.9)	27 (18.1)	

Note: *n* = numbers of participants who responded to each item. Chi-squared (Chi-Sq) test for categorical variables and independent *t*-test for continuous variable.

**Table 2 vaccines-10-01640-t002:** COVID-19 vaccination-related factors by vaccination status.

Variables	Non-Vaccinated COVID-19 Patients *n* (%)	Vaccinated COVID-19 Patients *n* (%)	*p*-Value
**Family or friends received vaccines (*n* = 385)**			
No	68 (87.2)	10 (12.8)	0.344
Yes	254 (82.7)	53 (17.3)	
**Received information about COVID-19 vaccines (*n* = 385)**			
No	23 (79.3)	6 (20.7)	0.525
Yes	299 (84.0)	57 (16.0)	
**Information sources (*n* = 385)**			
No information received	14 (100)	0 (0.0)	0.093
Newspaper/news channel	196 (83.1)	40 (16.9)	
Friends or family	80 (80.8)	19 (19.2)	
Social media	32 (88.9)	4 (11.1)	
**Willing to receive a vaccine (*n* = 322)**			
No	128 (39.8)	-	-
Yes	194 (60.2)	-	-
**Trust information shared on social media about the safety and efficacy of COVID-19 vaccines (*n* = 354)**			
Yes	120 (85.7)	20 (14.3)	0.235
No	173 (80.8)	41 (19.2)	

Note: *n* = numbers of participants who responded to each item. *p*-values were obtained by Chi-Sq test.

**Table 3 vaccines-10-01640-t003:** Number of items and respondents, reliability, and score distributions by domain included in the vaccine hesitancy questionnaire.

Questionnaire Domains	Number of Items	Scale (Possible Range)	Number of COVID-19 Patients	Mean (SD)	Skewness	Percent Variance	Reliability Coefficients Cronbach’s Alpha
Hesitancy in receiving COVID-19 vaccine	3	Strongly agree to Strongly disagree (1–5)	385	9.1 (5.1)	−0.119	76.4	0.91
Trust in COVID-19 vaccine	8	Strongly agree to Strongly disagree (1–5)	385	23.5 (13.9)	0.216	9.4	0.97
**Total**	**11**	Strongly agree to Strongly disagree (1–5)	**385**	**32.6 (17.9)**	**0.158**	**85.8**	**0.97**

**Table 4 vaccines-10-01640-t004:** Demographics of unvaccinated patients by their willingness to receive their vaccine. (n = 322).

Variables	Not willing to receive *n* (%)	Willing to receive *n* (%)	*p*-Value
**Age, mean (SD)**	50.5 (14.1)	44.9 (11.3)	0.001
**Age Groups**			
20–29	11 (33.3)	22 (66.7)	0.001
30–39	14 (31.1)	31 (68.9)	
40–49	40 (36.0)	71 (64.0)	
50–59	27 (33.3)	54 (66.7)	
60 or above	36 (69.2)	16 (30.8)	
**Gender**			
Male	75 (39.7)	114 (60.3)	0.512
Female	53 (40.2)	79 (59.8)	
**Education**			
No formal education	46 (63.0)	27 (37.0)	0.001
Primary	30 (48.4)	32 (51.6)	
Secondary	36 (30.8)	81 (69.2)	
Diploma/Graduation	16 (22.9)	54 (77.1)	
**Occupation**			
Unemployed	34 (65.4)	18 (34.6)	0.001
Self-employed	31 (34.1)	60 (65.9)	
Employed	28 (27.2)	75 (72.8)	
Retired	35 (46.7)	40 (53.3)	
**Marital status**			
Unmarried/never married	10 (34.5)	19 (65.5)	0.003
Married	80 (35.2)	147 (64.8)	
Separated/divorced/widowed	38 (58.5)	27 (41.5)	
**Geographical location**			
Urban	64 (33.0)	130 (67.0)	0.007
Sub-urban	25 (52.1)	23 (47.9)	
Rural	39 (49.4)	40 (50.6)	
**Long-standing illness**			
No	70 (35.2)	129 (64.8)	0.019
Yes	58 (47.5)	64 (52.5)	

Note: *n* = number of participants who responded to each item. Chi-Sq test for categorical variables and independent *t*-test for continuous variable.

**Table 5 vaccines-10-01640-t005:** Multivariate logistic regression examining the association of patient-related factors with vaccine acceptance and hesitancy.

	Not Received Vaccine (Referent)	Received COVID-19 Vaccine		Agreement—No or Lower Hesitancy (Referent)	Disagreement—Higher Hesitancy	
		OR (95% CI)	*p*-Value		OR (95% CI)	*p*-Value
**Age categories**						
20–29	1.0	0.10 (0.01–0.72)	0.001	1.0	0.56 (0.10–3.03)	0.501
30–39	1.0	0.10 (0.14–0.85)	0.021	1.0	1.53 (0.33–7.18)	0.588
40–49	1.0	0.26 (0.15–0.96)	0.040	1.0	1.22 (0.32–4.38)	0.770
50–59	1.0	0.31 (0.10–1.02)	0.053	1.0	2.28 (0.64–8.17)	0.204
**Gender**						
Male	1.0	0.54 (0.30–0.96)	0.037	1.0	3.56 (1.67–7.59)	0.001
**No family members received the vaccine**	1.0	0.55 (0.25–1.24)	0.152	1.0	1.06 (0.34–3.29)	0.915
**Not willing to receive the COVID-19 vaccine**	-	-	-	1.0	53.45 (23.78–120.16)	0.001
**Marital status**						
Unmarried	1.0	4.82 (0.82–28.41)	0.083	1.0	0.66 (0.14–3.07)	0.597
Married	1.0	3.51 (1.33–9.22)	0.011	1.0	1.23 (0.44–3.39)	0.687
**Education**						
No formal education	1.0	0.60 (0.19–1.89)	0.387	1.0	5.26 (1.85–14.97)	0.002
Primary education	1.0	1.03 (0.38–2.66)	0.996	1.0	2.59 (0.92–7.56)	0.070
Secondary education	1.0	1.33 (0.58–3.07)	0.499	1.0	1.99 (0.80–4.91)	0.136
**Geographical location**						
Urban	1.0	3.16 (1.27–7.87)	0.013	1.0	2.14 (0.88–5.20)	0.095
Sub-urban	1.0	3.88 (1.35–11.15)	0.012	1.0	2.05 (0.63–6.65)	0.233

Note: Variable(s) entered in the backward stepwise model: age, gender, education, occupation, marital status, geographical location, long-standing illness, received information about the vaccine, family members received the vaccine, and overall hesitancy. Reference groups: >60 years; female; family members received vaccine; planning to receive COVID-19 vaccine; separated/divorced/widowed; diploma/graduation or above; rural.

## Data Availability

The datasets generated during and/or analyzed during the current study are available from the corresponding author on reasonable request.

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
