# Peer review of "Vaccine Acceptance and Hesitancy among Hospitalized COVID-19 Patients in Punjab, Pakistan"

_vaccines, 2022, doi:10.3390/vaccines10101640_

Round 1

Reviewer 1 Report

Table1: The authors calculated the column percentages for the variables. Usually, we see the percentage of people by the independent variable. For example, gender, how many men are vaccinated and how many men are not vaccinated. The present results show, among the vaccinated, how many were men and women. Suggest giving percentages according to the independent variable selected. We would like to know how many (what %) were vaccinated and what % were not vaccinated in a particular age group. Suggest correcting accordingly.

Regarding age groups, suggest taking equal interval age groups like 20-29; 30-39 etc., rather than quartile age groups.

Similarly, in Table 2, row percentages are to be given except for the variable, ‘planning to receive a vaccine’. We always try to assess based on the independent variable.

Table 5, I suggest including the proportion of reception of covid vaccination in  each independent variable category (like by age group, gender, marital status, education, geographic location). Also, authors may write about selecting these five independent variables for final regression. These are the only variables on which data were available? There were other variables like access to information, source of information, etc. Are these variables considered for regression? If not, how were they excluded? Are there any criteria followed to identify/select independent variables for final regression analysis?

Author Response

Open review

English language and style

( ) Extensive editing of English language and style required
( ) Moderate English changes required
(x) English language and style are fine/minor spell check required
( ) I don't feel qualified to judge about the English language and style

Yes

Can be improved

Must be improved

Not applicable

Does the introduction provide sufficient background and include all relevant references?

(x)

( )

( )

( )

Are all the cited references relevant to the research?

(x)

( )

( )

( )

Is the research design appropriate?

( )

(x)

( )

( )

Are the methods adequately described?

( )

( )

(x)

( )

Are the results clearly presented?

( )

( )

(x)

( )

Are the conclusions supported by the results?

( )

( )

( )

(x)

Comments and Suggestions for Authors

Comment

Table1: The authors calculated the column percentages for the variables. Usually, we see the percentage of people by the independent variable. For example, gender, how many men are vaccinated and how many men are not vaccinated. The present results show, among the vaccinated, how many were men and women. Suggest giving percentages according to the independent variable selected. We would like to know how many (what %) were vaccinated and what % were not vaccinated in a particular age group. Suggest correcting accordingly.

Author response

Thank you for the comment. We have modified the data presentation of table 1 and presented as per your suggestions. We are hoping it will be accepted now.

Comment

Regarding age groups, suggest taking equal interval age groups like 20-29; 30-39 etc., rather than quartile age groups.

Author response

Thank you for the suggestion. In order to get appropriate population in respective groups this age category used. But we have changed this to what you have suggested in Table 1.

Comment

Similarly, in Table 2, row percentages are to be given except for the variable, ‘planning to receive a vaccine’. We always try to assess based on the independent variable.

Author response

Thank you for your suggestion. We have changed the format to what you have suggested. Earlier we presented both row and column numbers/ proportions for better understanding of dependent and independent variables. For example, in table 1, you can tell how many people aged 21-40 were vaccinated or not vaccinated as well as total number of people who were aged 21-40.

Comment 

Table 5, I suggest including the proportion of reception of covid vaccination in each independent variable category (like by age group, gender, marital status, education, geographic location). Also, authors may write about selecting these five independent variables for final regression. These are the only variables on which data were available? There were other variables like access to information, source of information, etc. Are these variables considered for regression? If not, how were they excluded? Are there any criteria followed to identify/select independent variables for final regression analysis?

Author response

Selecting covariates/ confounders/ mediators is always a debatable issue. Selecting everything is not appropriate as this approach may lead to overadjustment of regression models. There are various approaches to selecting appropriate and relevant covariates in regression models. We have used a multivariate logistic regression with a backward stepwise model approach (as mentioned in our methods). Table 5 represents our approach to regression analysis, presenting important variables, selected using the backward stepwise model approach

Reviewer 2 Report

This paper on vaccine hesitancy could have provided an important piece of the vaccine hesitancy puzzle because it focuses on Pakistan, for which there is not much research about this topic.  However, the paper makes no attempt to review the extensive body of literature on this topic.  Specifically, many studies find that hesitancy is driving by issues such as religion, culture, political preferences, vaccination schedule, mode of administration reliability and cost concerns and individual group factors such as risk/benefit analysis and prior experiences.  These are the kinds of questions that the authors should have asked respondents to understand the drivers of vaccine hesitancy.  It's not enough to know that people are hesitant.  It's vital to know why they are hesitant so health communicators can design interventions to deal with these concerns.  

In addition to this very important issue, the data are presented in a very confusing manner.  The tables basically provide the raw data for readers to look at.  Summarizing these data into more consumable indicators provides a much more coherent picture of what the data are telling us.  In addition, Structural Equation Modeling would be a more appropriate method to use in analyzing the data, particularly if more data were collected to better understand the causes of hesitancy.  

Unfortunately, this study provides little useful information about the hesitancy issue.  Thus, I recommend rejecting this manuscript for publication.  

Author Response

Open Review

English language and style

( ) Extensive editing of English language and style required
( ) Moderate English changes required
(x) English language and style are fine/minor spell check required
( ) I don't feel qualified to judge about the English language and style

Yes

Can be improved

Must be improved

Not applicable

Does the introduction provide sufficient background and include all relevant references?

( )

( )

(x)

( )

Are all the cited references relevant to the research?

( )

( )

(x)

( )

Is the research design appropriate?

( )

( )

(x)

( )

Are the methods adequately described?

(x)

( )

( )

( )

Are the results clearly presented?

(x)

( )

( )

( )

Are the conclusions supported by the results?

( )

( )

(x)

( )

Comments and Suggestions for Authors

Comment

This paper on vaccine hesitancy could have provided an important piece of the vaccine hesitancy puzzle because it focuses on Pakistan, for which there is not much research about this topic.  However, the paper makes no attempt to review the extensive body of literature on this topic.  Specifically, many studies find that hesitancy is driving by issues such as religion, culture, political preferences, vaccination schedule, mode of administration reliability and cost concerns and individual group factors such as risk/benefit analysis and prior experiences. These are the kinds of questions that the authors should have asked respondents to understand the drivers of vaccine hesitancy. It's not enough to know that people are hesitant.  It's vital to know why they are hesitant so health communicators can design interventions to deal with these concerns.  

Author response

Thank you for your comment. We have added more information in our introduction. It is difficult to perform and include a complete literature review in introduction. We have introduced the topic and relevant background information in the introduction before presenting the objectives.

We have designed the questionnaire keeping the target population in mind. Population visiting these public health facilities are mostly from under-privileged groups with secondary or less education, asking too many questions discourage them from participating. Besides religion, culture and politics are important factors but these are sensitive topics in Pakistan, discouraging people to talk about it openly. We have developed enough number of questions for them to understand and answer. Some factors like cost, is not relevant because these vaccines were offered free of cost at public facilities. Our focus was more on the aspect of ‘trust’ to explore hesitancy

Comment

In addition to this very important issue, the data are presented in a very confusing manner.  The tables basically provide the raw data for readers to look at.  Summarizing these data into more consumable indicators provides a much more coherent picture of what the data are telling us.  In addition, Structural Equation Modeling would be a more appropriate method to use in analyzing the data, particularly if more data were collected to better understand the causes of hesitancy. 

Author response

We have revised the first two tables based on the reviewer 1 comments.

Tables 1 and 2 presents study data by vaccination status. Table 3 présents questionnaires structure and domain.

We have added a new figure (Fig 1) before Table 4, which presents the patient-related factors associated with the willingness of unvaccinated participants to receive COVID-19 vaccine.

Table 4 is an important Table presenting willingness of unvaccinated participants to receive COVID vaccines in future. This table also summarises how many of them are reluctant to accept vaccine in future.

Before Table 5, we have added Figure 2 to explain the patient-related factors related to hesitancy score. This provides a good background to Table 5

We would be happy to recruit more patients but due to reduced cases of COVID-19 in Pakistan and limited cases in hospitals, it is difficult to recruit more people at this stage. We have recruit more than 350 patients, and we think adding more would not change the results greatly.

We have used multivariate logistic models with backward stepwise approach to examine the association which is widely used in epidemiological studies. We’ll definitely consider SEM in our future studies.

Reviewer 3 Report

Review for: "Vaccine Acceptance and Hesitancy among hospitalized COVID-19 Patients in Punjab, Pakistan": This is an interesting manuscript on a vibrant and important subject. Nevertheless, the following comments should be taken into account before further consideration at Games.

1) I would encourage the authors to extend the abstract more with the key results. As it is, the abstract is a little thin and does not quite convey the interesting results that follow in the main paper. The focus should be more on the key new results. Currently, the abstract is quite verbose and the main messages dilute in such a style.

2) The presentation is not in keeping with an interdisciplinary readership. Too much technical details is presented without much guidance of the reader through what is shown and why. This needs improvement for better clarify of the presentation.

3) It would also improve the paper if the table captions would be made more self contained. In addition to what is shown, one could also consider a sentence or two saying what is the main message of the data in the tables.

4) The introduction should be improved by referring also to closely related recent research regarding vaccination and hesitancy, such as: Risk assessment of COVID-19 epidemic resurgence in relation to SARS-CoV-2 variants and vaccination passes, T. Krüger et al., Commun. Med. 2, 23 (2022) and Optimal governance and implementation of vaccination programmes to contain the COVID-19 pandemic, Mahendra Piraveenan, et al., R. Soc. Open Sci. 8, 210429 (2021). These are recent, related, and relevant for the questions the authors are addressing in their manuscript.

5) Also, it would be very useful if the authors would make their data and source code available as supplementary material. This would promote the usage of the proposed data and allow also others to take advantage of this research, and also to allow them to reproduce the results.

6) I am missing a comprehensive discussion of the limitations of this research and the conclusions as the data do not take into account any actual network of contacts, which have been emphasized so much for best vaccination practices. The authors need to discuss this point carefully and acknowledge the lack of taking this important aspect into account.

7) Some references contain errors, missing or incorrect information, and inconsistent formatting. It is difficult to give credit to research if such elementary aspects of the work are not error free. References should thus be corrected with the best care.

If a revision will be granted, I will be happy to review the manuscript again.

Author Response

Open Review

English language and style

( ) Extensive editing of English language and style required
(x) Moderate English changes required
( ) English language and style are fine/minor spell check required
( ) I don't feel qualified to judge about the English language and style

Yes

Can be improved

Must be improved

Not applicable

Does the introduction provide sufficient background and include all relevant references?

( )

(x)

( )

( )

Are all the cited references relevant to the research?

(x)

( )

( )

( )

Is the research design appropriate?

(x)

( )

( )

( )

Are the methods adequately described?

( )

(x)

( )

( )

Are the results clearly presented?

( )

(x)

( )

( )

Are the conclusions supported by the results?

( )

(x)

( )

( )

Comments and Suggestions for Authors

Review for: "Vaccine Acceptance and Hesitancy among hospitalized COVID-19 Patients in Punjab, Pakistan": This is an interesting manuscript on a vibrant and important subject. Nevertheless, the following comments should be taken into account before further consideration at Games.

 Comment

1) I would encourage the authors to extend the abstract more with the key results. As it is, the abstract is a little thin and does not quite convey the interesting results that follow in the main paper. The focus should be more on the key new results. Currently, the abstract is quite verbose and the main messages dilute in such a style.

Author response

Thank you for the comment. We have updated the abstract per your comment and incorporated the main findings of the study. We are hoping it will be acceptable now.

 Comment

2) The presentation is not in keeping with an interdisciplinary readership. Too much technical details is presented without much guidance of the reader through what is shown and why. This needs improvement for better clarify of the presentation.

Author response

We have added two figures before the main analyses to explain the purpose better. We have added more descriptions in each table and figure and the methods. We also changed the presentation of results in Tables 1 and 2 based on reviewer 1 suggestion.

Comment

3) It would also improve the paper if the table captions would be made more self-contained. In addition to what is shown, one could also consider a sentence or two saying what is the main message of the data in the tables.

Author response

We have added a description to each table and figure. All information about statistical tests also added in the footnote

Comment 

4) The introduction should be improved by referring also to closely related recent research regarding vaccination and hesitancy, such as: Risk assessment of COVID-19 epidemic resurgence in relation to SARS-CoV-2 variants and vaccination passes, T. Krüger et al., Commun. Med. 2, 23 (2022) and Optimal governance and implementation of vaccination programmes to contain the COVID-19 pandemic, Mahendra Piraveenan, et al., R. Soc. Open Sci. 8, 210429 (2021). These are recent, related, and relevant for the questions the authors are addressing in their manuscript.

Author response

Thank you for the comment. We have updated the introduction section and incorporated recent studies about vaccine hesitancy. We also tried to present the regional issues along with other factors.

Comment 

5) Also, it would be very useful if the authors would make their data and source code available as supplementary material. This would promote the usage of the proposed data and allow also others to take advantage of this research and also to allow them to reproduce the results.

Author response

Thank you for the suggestion. The deidentified data set of the current manuscript is available as a ‘supplementary file’ for the kind review of readers.

 Comment

6) I am missing a comprehensive discussion of the limitations of this research and the conclusions as the data do not take into account any actual network of contacts, which have been emphasized so much for best vaccination practices. The authors need to discuss this point carefully and acknowledge the lack of taking this important aspect into account.

Author response

We did include a comprehensive paragraph before the conclusion. We have further extended this section to include more limitations

Comment

7) Some references contain errors, missing or incorrect information, and inconsistent formatting. It is difficult to give credit to research if such elementary aspects of the work are not error free. References should thus be corrected with the best care.

Author response

Thank you. We have corrected our references section

Round 2

Reviewer 2 Report

The manuscript is slightly improved over the original.  The authors stated that doing even a minimal literature review would be too difficult so it was not attempted.  A quick search in Google Scholar will reveal hundreds of articles on vaccine hesitancy including many summary articles.  If the authors included even one or two of those, it would be a big improvement.  Otherwise the article is fine to publish.  

Author Response

Comment

The manuscript is slightly improved over the original.  The authors stated that doing even a minimal literature review would be too difficult so it was not attempted.  A quick search in Google Scholar will reveal hundreds of articles on vaccine hesitancy including many summary articles.  If the authors included even one or two of those, it would be a big improvement.  Otherwise, the article is fine to publish.

Author response

Thank you again for your comment. As per your comment, we have updated introduction section (From line 99-103) by incorporating factors associated with COVID-19 vaccines hesitancy in Pakistan. We are hoping that it will be acceptable now.

Reviewer 3 Report

The authors have revised their manuscript comprehensively and with love to detail. I warmly recommend publication in present form.

Author Response

Comment

The authors have revised their manuscript comprehensively and with love to detail. I warmly recommend publication in present form.

Author response

Dear reviewer thank you for your time and endorsement for publication of our manuscript.
